# Affective Attitudes in the Face of the COVID-19 Pandemic: The Dynamics of Negative Emotions and a Sense of Threat in Poles in the First Wave of the Pandemic

**DOI:** 10.3390/ijerph192013497

**Published:** 2022-10-19

**Authors:** Anna Szuster, Miroslawa Huflejt-Łukasik, Dorota Karwowska, Maciej Pastwa, Zuzanna Laszczkowska, Kamil K. Imbir

**Affiliations:** Faculty of Psychology, University of Warsaw, 00-927 Warszawa, Poland

**Keywords:** negative emotions, threat, physical distance, COVID-19

## Abstract

For millions of people, the COVID-19 pandemic situation and its accompanying restrictions have been a source of threat and confrontation with negative emotions. The pandemic’s universal and long-term character, as well as the ensuing drastic limitation of control over one’s life, have made it necessary to work out adaptive strategies that would reduce negative experiences and eventually lead to the restoration of well-being. The aim of this research was to identify strategies that people use in response to a long-term threat that restore affective balance and a subjective sense of security. We registered selected manifestations of affective reactions to the pandemic situation. The researchers focused on the dynamics of changes in the areas of (1) experienced negative emotions (asked in an indirect way) and (2) a subjective feeling of threat regarding the pandemic (in three different contexts: Poland, Europe, and worldwide) during the first phase of the pandemic in Poland. It was expected that both the negative emotions and the sense of threat would decrease with time. In addition, it was anticipated that the physical distance would modify the assessment of the situation as threatening depending on the geographical proximity: in Poland, Europe, and worldwide. We used the mixed quasi-experimental design in the series of four studies conducted by Internet in March, May, June, and July 2020. The intensity of negative emotions and the sense of threat caused by the pandemic situation in Poland, Europe, and worldwide were measured. Despite the objective number of confirmed COVID-19 cases during each of the stages of the study, both the intensity of emotions attributed by participants as well as the feeling of threat were found to have decreased. In addition, surprisingly, a reversed effect of the distance was revealed: namely, a sense of threat experienced towards distant locations (Europe and the world) was found to be more acute when compared with the threat experienced in Poland. The obtained results are interpreted as a manifestation of adaptive perception of the threat that lies beyond one’s control, which takes the form of unconscious, biased distortions: unrealistic optimism. The decrease in the intensity of negative emotions explains unrealistic absolute optimism, while the perception of the situation in Poland as less threatening than in Europe and around the world is predicted by unrealistic comparative optimism.

## 1. Introduction

The coronavirus disease (COVID-19) pandemic represents a serious threat to millions of people around the world [1]. The recorded numbers of deaths, serious disease courses, and infections clearly show the deadly power of this disease. Confrontation with necessity to make changes in almost all aspects of human functioning, the unpredictability of events, loss of control over one’s course of life, the need to isolate and limit contacts, and fear for survival are all sources of strong affective reactions. Predicting affective consequences of the pandemic is a task that goes far beyond the cognitive capabilities of not only individuals or groups but also experts and governments. One’s response to the long-term prospect of having to remain in such a situation is to work out adaptive strategies of coping with the sense of threat and one’s negative emotions.

The aim of the study presented in this article was to identify strategies that people use in response to the need for long-term confrontation with a growing threat. We focused on the dynamics of the experienced affective states at different times of the first outbreak of the pandemic: the intensity of the negative emotions expected to be experienced in a population (as an indirect measure of emotions experienced by individuals) and a sense of threat caused by the pandemic situation depending on the distance from the self and in locations where geographical distances from the self varied (Poland, Europe, and worldwide). The accumulated knowledge about the dynamics of affective states in the face of a threatening situation provides information on how people cope with a long-term danger. This, in turn, may prove helpful in devising more effective prophylactic and therapeutic measures in response to the psychological consequences of the pandemic.

### 1.1. Negative Emotions and a Sense of Threat: The Affective Response to the Pandemic Situation

Emotions have arisen due to the adaptive role they play in solving life tasks understood (also) as universal elements of human experience, such as achievements, losses, and frustrations. Emotion triggers a course of action that helps cope with important, repetitive situations from the perspective of human goals [2]. Certain unchangeable characteristics are typical of emotion-generated reactions. Both emotions and their changes occur quickly. The speed at which emotions appear causes a person to subjectively perceive them as a kind of experience that happens rather than as something that he/she chooses. This holds true also with regard to one’s reaction to a situation of such threat as the pandemic. Experiencing negative emotions, including fear, sadness, insecurity, or disgust, is typical for most people who are aware of the epidemic threat [3]. They perform the functions of mobilisation [4], information [5], and communication [6]; they trigger the processes of self-regulation [7] and influence social perception; and sometimes they are a source of distortion.

A sense of threat in relation to the experienced stress is perceived in transactional categories [8]. The participant’s subjective assessment is decisive in perceiving a certain relation as stressful. If a situation is viewed as burdensome or exceeding the person’s resources and threatening his or her well-being, and, consequently, requiring an adaptive effort, it is defined as stressful and threat-inducing [9].

Though the sense of threat is inevitably associated with experiencing negative emotions, it constitutes a different specific response to the pandemic situation. A pandemic caused by a new pathogen, together with the uncontrolled spread of infections, creates an emergency situation. Affective states and the accompanying high-risk-induced arousal increase the tendency to selectively process information. Empirical data indicate that sudden change and unexpected situations such as being confronted with COVID-19 are perceived as a threat [10] and elicit negative emotions such as sadness, anxiety, anger, and hostility, and, in turn, that anxiety and anger/hostility have an influence on the perceived threat caused by COVID-19 [11,12,13].

The majority of studies conducted during the COVID-19 pandemic in various countries systematically have shown that the changes caused by the situation connected with COVID-19 are associated not only with negative emotions, but also an increase in psychopathological symptoms, including psychosomatics and the perceived threat associated with it [14,15,16,17]. It was found that tolerance of uncertainty is a mediator between the perceived COVID-19 threat and psychosomatic complaints. Fear during COVID-19 also manifests in feelings of loneliness, uncertainty, anxiety, or even panic [18].

Subjective experience of fear, worry, and threat is associated with mental health and symptoms of depression and anxiety. Additionally, social vulnerability (e.g., women, families with children) is connected with higher sensitivity and more intensive negative feelings about COVID-19 and also a higher risk of anxiety and depressive symptoms. A long-term confrontation with the threat generated by the epidemic not only aggravates depression symptoms (particularly in certain age groups) and is conducive to risk behaviours but also serves as a background for increased conspiratorial thinking [19]. In addition, semantic network analysis indicates a clear trend associated with emotion (negative emotions had their highest scores in mid-to-late March, when the WHO announced the pandemic status of COVID-19) [20].

Research has demonstrated that both frontline medical staff and the general public are experiencing a range of psychological problems of stress-related disturbances, sense of threat, anxiety, and depression [17,21,22,23,24,25]. The image of the psychological impact of quarantine, which emerges from the review of the literature, shows the effects in the form of post-traumatic stress disorder, as well as negative emotions such as confusion and anger [22]. Longer quarantine brings infection fears, frustration, boredom, and stigma. The research points to a fairly coherent pattern of affective reactions to the pandemic situation in most societies, where negative affect (including emotions) and a sense of threat prevail. It is already known that anxiety related to a pandemic intensifies clinical symptoms of phobias, social anxiety, and depression [26], and anxiety and threat are indicated as the key problems of the pandemic.

### 1.2. Current Studies

Social perception is determined by psychological criteria, the sources of which are the current interests and needs of the subject [27]. Moreover, the perception of affective states and the sense of threat is modified by such “psychological reasons” that produce an image of reality that is adaptive to the subject [28]. For example, within a month from the date when the pandemic was officially declared in Poland [29], the knowledge about ignoring the threat in the form of unrealistic optimism could be used to introduce specific measures aimed at promoting safe behaviours (distance, masks).

The research presented in this article pursued two goals. The first was to verify predictions regarding the dynamics of affective states (negative emotions and a subjective feeling of threat) in the face of the first outbreak of the pandemic in Poland. The results of some of the research point to a complex array of emotions experienced in the long-term contact with a threatening situation. Many of them show, in the longer perspective, dynamics of people’s experiences during COVID-19. For example, in Germany, despite the objective danger in the period of December 2019 through March 2020, average life satisfaction and the positive and negative emotions experienced by people did not change [30], and decreased between March and May 2020.

Another study showed a negative affect reduction in the subsequent months of the pandemic: a decreased sense of danger between March and June followed by its increase in the following months in Great Britain [31], whereas in the US a decrease between April and June and a subsequent increase were recorded [32]. In Austria, a significantly decreased sense of fear and stress and a rise in the quality of life were observed between April and September 2020 [33].

Research supports, therefore, that emotional reactions to COVID-19 were changing. In the first period, fear, anger, and anxiety prevailed, but later, calm, hope, leisure, and solidarity with others dominated [34]. Even during the first week of lockdown due to the COVID-19 pandemic, Poles experienced positive emotions—relaxation and happiness—more frequently and more intensely than negative emotions, such as anger, anxiety, and sadness (fear episodes were even less frequent) [35].

These results allow one to expect that the intensity of the experienced affect in the form of negative emotions and a sense of threat would decrease in subsequent phases of the pandemic. The rationale for this expectation is the regulatory function of mechanisms that increase the chance of adapting to the conditions of life marked with a psychological burden. Two hypotheses concerning the dynamics of negative emotions and the feeling of threat were formulated.

**Hypothesis** **1** **(H1).**
*The intensity of negative emotions changes with time: it declines in subsequent months of the pandemic compared with the intensity recorded at the onset.*


**Hypothesis** **2** **(H2).**
*The intensity of the sense of threat changes with time: it decreases in subsequent months of the pandemic compared with the intensity recorded at the onset.*


The second aim of the current research was to check whether physical distance affects the assessment of the pandemic situation. Despite the global character of the pandemic and its unified manifestations (in the form of infections and deaths), much of the data indicate that subjective assessments depend also on the physical [34] or social distance perceived by the subject [36,37]. Thus, while designing the present study, we focused on how the pandemic threat was perceived by Poles depending on place: in Poland, Europe, and worldwide.

It was therefore anticipated that the assessment of the pandemic situation as threatening would differ depending on the location. One’s own actual place of living during the pandemic is perceived as less threatening in comparison with more distant geographical locations. The results of studies in the field of social psychology, as well as ones conducted during the pandemic, appear to support such predictions.

The sense of danger influences the perception of reality, assessment, and prediction of future events. People tend to ignore important data, which results in errors in probability estimates and event judgments [28]. Physical distance in such a threatening situation is a factor that promotes distortion. The same situation in distant places is assessed differently than when it concerns the place where we live. For example, physical distance was also the factor modifying subjective assessment of the car accident outcome: information confirming that the accident occurred at a distant location decreased negative emotions [38].

Some research [39] indicates that perceiving the situation as threatening and dangerous depends, among other things, on the physical distance: the greater the distance, the lower the arousal and subjectively lower the sense of assessed threat. Persons remaining within a close distance are favoured (e.g., by way of help offered) compared with those with the same needs and of the same status yet remaining at a more distant location [40,41]. These results confirm that the social and physical distance influences the assessment of a situation, including how threatening it is.

Research conducted during the pandemic verifies this regulatory role of distance with regard to persons perceived, to varying degrees, as close. Jordan et al. [42] found that focusing on “your community” promotes intentions to engage in preventive behaviours compared to the “baseline”. In addition, results of another study confirm that focusing on “your community” promotes intentions to wear a face mask compared with focusing on “my country” and “my family”, also compared to the baseline [1]. Likewise, priming based on the categories defining the immediate social and physical space of the subject (“your community”) best predicts the prophylactic intentions in the situation of the pandemic threat.

Thus, proximity (short distance) intensifies activities aimed at reducing the risk of infection. This, in turn, justifies the feeling of a smaller threat in a situation that is directly present. This distinct status of a “close” location justifies the prediction contained in hypothesis 2b, presented below, regarding the specificity of the sense of threat depending on location.

**Hypothesis** **2b** **(H2b).**
*Irrespective of the passing of time during the first wave of the pandemic, subjective assessment of the pandemic situation as threatening shall be lower with regard to the situation of the pandemic in Poland compared with the situation in Europe and around the world.*


## 2. Method

We applied a mixed quasi-experimental design: between subjects at stage 1 vs. 2, 3, and 4 and within subjects at stages 2, 3, and 4.

### 2.1. Participants

The first stage of the study was conducted on a different sample than the other stages. The three consecutive stages of the study (stages 2–4) used the same pool of responders. However, the number of participants decreased from stage to stage. In the first stage, the participants were volunteers recruited to the study from groups and forums on social media (e.g., student groups or groups associated with the inhabitants of certain cities and towns). The survey was published using Qualtrics services. Participants could enter the survey only once, and they did not receive remuneration for participation. In the following three stages of the study, the participants were recruited by the Ariadna research panel. The panel recruits participants using many methods, such as social media, mailings, and the snowball method. The participants in stages 2 to 4 received remuneration for participating in the study, namely points in the Ariadna panel, which could have been subsequently exchanged for products in the system provided by the research panel. The participants in stages 2 to 4 could also enter the survey only once.

All participants who did not respond to even one of the questions were excluded from the study. The following description concerns the final sample, the data from which were analysed in the study. The sample in the first stage of the study consisted of 1243 participants, including 1022 women (82.2%), 216 men (17.4%), and 5 research subjects who described their sex as other (0.4%). Age in this group ranged between 14 and 78 (M = 34.52; SD = 13). The second stage of the study was conducted on a group of 1130 participants, 569 of whom were women (50.4%) and 561 men (49.6%). The number of research subjects diminished in succeeding stages. In the third stage, the group consisted of 971 participants, with 473 women (48.7%) and 498 men (51.3%). In the fourth stage, the sample consisted of 818 research subjects, with 387 women (47.3%) and 431 men (52.7%). These data are summarised in Table 1 below.

Participants in stages 2 to 4 were aged between 18 and 85 years old (M = 44.53, SD = 15.84). A total of 328 participants (29.03%) lived in big cities with over 100,000 inhabitants, and 802 (70.97%) lived in villages, small towns, or cities with populations under 100,000. Respondents were at different stages of education, with 45.75% (517 research participants) having completed higher education. Each of the four stages captured the sample in a different situation regarding exposure to the virus and the threat it implied, as they were taken during different stages of coronavirus pandemic development, described in Section 2.5., Data concerning the epidemic situation in Poland during the study.

### 2.2. Design 

We used a mixed quasi-experimental scheme. The intensification of negative emotions and the sense of threat at three different places (Poland vs. Europe vs. worldwide) were measured in four stages of the study. All other measures were also conducted in four stages. 

### 2.3. Materials 

#### 2.3.1. Negative Emotions 

The participants were asked about emotions with the following prompt: “Assess to which extent emotions given below cause people to experience threat in the current situation”. We used the indirect measurement of intensity of negative emotions. We asked about the emotions experienced by people in general, as we expected that a direct question about a particular participant’s emotions would activate a defensive mechanism, causing participants to answer conservatively and in a socially acceptable way. The results of a pilot study comparing the direct and indirect methods of measuring negative emotions are presented in the results section of this paper. In the first stage of the main study, the assessments were done on a scale from 1 (“to a small extent”) to 100 (“to a high extent”). The scale had the form of sliders, with the start position set in the middle. The participants did not see the height of the assessment (number), and there were no grid lines near the slider. The emotions were suffering, helplessness, frustration, breakdown, terror, bitterness, aversion, disgust, abhorrence, repulsion, humiliation, shame, embarrassment, disappointment, disillusionment, sadness, sorrow, depression, envy, and contempt. The emotions were presented in a random order for each participant. From the second to the fourth stage, the assessments were done on a scale from 1 (“to a small extent”) to 7 (“to a high extent”). The same 20 emotions were presented in a random order for each participant. For the scale measuring negative emotions, the Cronbach’s α had the value of α = 0.89 in the first stage of the study, α = 0.96 in the second stage, α = 0.97 in the third stage, and α = 0.98 in the fourth stage. The particular emotions were picked in order to cover a large span of negative affective states that could be experienced, both automatic (deriving from homeostasis) and reflective (based on self-standards and/or social cognition) [43]. The words representing negative emotions measured in this study were previously used as experimental stimuli evoking negative incidental affect [44].

#### 2.3.2. Sense of Threat at Different Distances

The measurement procedure commonly used in numerous studies on the influence of social [36,37] and physical [40,41] distance on various manifestations of attitudes was used. We used an indirect measurement of the sense of threat at different distances, with three questions to measure this variable. The first question was “To what extent do you feel threatened by the epidemic situation in Poland?” The second was “To what extent do you feel threatened by the epidemic situation in Europe?” And the third was “To what extent do you feel threatened by the epidemic situation around the world?” All three questions were answered on a scale from 1 (“I feel threatened to a small extent”) to 7 (“I feel threatened to a large extent”) in all stages of the study. 

#### 2.3.3. Questions Regarding Participants’ Demographics

The last set of questions used in the study concerned the participants’ demographics. The participants were asked “Have you contracted the COVID-19 disease?” and “Has someone you know contracted the COVID-19 disease?” (this question was used only in the first stage of the study). They also declared their sex, their age, the size of their place of residence (village, town, city), and their education. At the end, the participants were asked whether they had a full-time job and whether, in the face of the pandemic, they had their payment assured.

### 2.4. Procedure 

The study design and procedure were approved by the bioethical committee of the Faculty of Psychology at the University of Warsaw. All procedures involving human participants were conducted in accordance with the ethical standards of the institutional and/or national research committee and with the 1964 Helsinki Declaration and its later amendments or comparable ethical standards. At each stage of the study, participants gave their consent by answering the question in the survey.

The study was conducted using an online questionnaire (Appendix A). There were four stages of the study, each including the same questions, as described above. It is important to note that as the spread of the pandemic proceeded, each stage of the study was conducted while there was an increasing number of noted coronavirus cases (see Section 2.5).

The respondents initially answered questions about negative emotions, then ones regarding the sense of threat at different distances (in Poland, Europe, and worldwide). The questions regarding participants’ demographics were answered at the end. All questions had to be answered in one session. The questionnaires, both in Polish and in English, can be found in the Appendix. The questionnaire was part of a larger research project regarding the COVID-19 pandemic conducted by the Faculty of Psychology at the University of Warsaw. Subjects received remuneration for participation in the project (except in the first stage of the study, in which participants took part voluntarily). The scheme of the study is presented in Figure 1.

### 2.5. Data Concerning the Epidemic Situation in Poland during the Study

The study consisted of four stages. The first stage took place between 19 and 24 March 2020. The second stage took place between 4 and 7 May. The third stage was conducted between 4 and 17 June. Finally, the last stage took place between 7 and 17 July. 

In order to provide a broader context for the data collected in our study, we report data explaining the social and political situation in Poland, Europe, and around the world, gathered by Be Communication, a public relations company. The information collected preceded each stage of our study by five to seven days, as the news and media reports would bring a change to one’s mood after a few days. The reported data included a few main areas regarding the pandemic situation worldwide and in Poland. 

The first aspect of the pandemic concerned statistical data such as confirmed cases and deaths in Poland, Europe, and worldwide. Those data were collected from the official WHO page (https://covid19.who.int/, accessed subsequently for each stage of the study on 12 March 2020—stage 1; 29 April 2020—stage 2; 29 May 2020—stage 3; 2 July 2020—stage 4) and the official page of the Polish Ministry of Health (https://www.gov.pl/web/koronawirus/pliki-archiwalne-dane-historyczne, accessed subsequently for each stage of the study on 12 March 2020—stage 1; 29 April 2020—stage 2; 29 May 2020—stage 3; 2 July 2020—stage 4). The second aspect of the pandemic included information about daily amendments in regulations and restrictions imposed by the Polish government. The third aspect regarded the political and economic situation in Poland as well as general social initiatives taken by citizens. The information was collected using the Google search engine and included the most known Polish news websites (the same websites were used each time). The information was taken from Internet sites such as onet.pl, gazetaprawna.pl, RMF24.pl, SuperBiz.SE.pl, Polska Times, Wirtualna Polska, TVN24, TVP Info, Radio Zet, Polityka.pl, PolskieRadio24.pl, Fakt.pl, and WiadomosciGazeta.pl.

The situation in Poland preceding the first stage of the study (March 2020) was as follows: At that time, all schools and universities had already been closed for a few days. A number of employers prepared to use the home office option. In general, media were discussing the global epidemic situation, with their main focus on Italy and France, as well as the rapid growth in the number of COVID-19 cases in those countries. Meanwhile, in Poland, the first victim of COVID-19 passed away. That same evening, 13 March, the Polish government introduced an epidemic emergency for the next two weeks, followed by new restrictions. The restrictions included border control and 14 days of quarantine for people entering Poland.

The situation preceding the second stage (May 2020) was as follows: Two months later, the number of people infected with coronavirus exceeded 14,000; the number of fatalities approached 700; and about 4000 people had recovered. The media were stressing information on the insufficient number of tests to detect the presence of the coronavirus (especially among physicians) and experts’ opinions that the number of infected may be between 40,000 and 50,000. Although for about two weeks all Poles had been obligated to wear masks, the most restrictive recommendations that locked people down in their homes had already been loosened: the ban on recreation in forests and parks was removed, and the number of people in shops had increased.

The situation preceding the third stage (June 2020) was as follows: The number of people infected with coronavirus exceeded 28,200; the number of fatalities approached 1200; and about 13,700 people had recovered. According to news reports and press articles, people paid less attention to coronavirus in spite of the fact that the number of infected was not decreasing. The coronavirus attracted less and less public attention, although the number of cases was not falling, and experts were not sure whether the peak of the incidence had been reached. The situation had worsened in several voivodeships, and there were large outbreaks of infections (mines, nursing homes, hospitals). At the same time, the so-called defrosting of the country took place. Almost all restrictions introduced to contain the pandemic had been lifted. Only wearing masks in shops, means of communication in closed spaces, and keeping social distance were obligatory, but fewer and fewer people respected these restrictions. Moreover, although the number of people infected with the coronavirus had almost doubled compared to the previous month, media commented that “society spontaneously announced the end of the pandemic”.

The situation preceding the third stage (July 2020) was as follows: Four months of the pandemic had passed. The number of people infected with coronavirus exceeded 38,700; the number of fatalities approached 1600; and about 28,500 people had recovered. The daily number of cases in Poland was still between 200 and 400. The press reported on outbreaks of infections at individual hospitals, workplaces, weddings, baptisms, and funerals, and even at holiday centres. The need for normalisation was strong everywhere. Fewer and fewer people cared about maintaining the sanitary restrictions. More and more people were without protective masks in stores and means of transport. The analysed period was a time strongly marked by politics and the second round of the presidential elections. The Prime Minister assured the public that the situation related to the coronavirus was stabilising and that a vaccine was on its way that would definitely solve the problem. Depending on the type of medium and the media source, information about the threat of coronavirus tended to be more and more contradictory.

### 2.6. Data Analysis

There were two independent samples. The data acquired in the first stage of the study were obtained from the first sample. The data acquired in stages 2–4 were obtained in a second sample of participants, who participated in three repeated measurements. Repeated measures analysis of variance was used to assess differences in the intensity of negative emotions between these three consecutive research stages. Two within-subject effects were analysed to verify our hypotheses, the effect of the research stage, and the effect of perspective (Poland, Europe, or worldwide). The differences between stage 1 and stages 2 to 4, which were used as baseline comparison values, were assessed with a Student’s *t*-test, with Bonferroni correction lowering the statistical significance threshold value to *p* < 0.017. There were three comparisons to make. The data acquired in stage 1 were compared to the data acquired in stages 2, 3, and 4. Therefore, we divided the conventional cut-off value for *p* by 3. The same procedure was applied to both dependent variables: the intensity of negative emotions and the sense of danger.

## 3. Results

### 3.1. Negative Emotions: Pilot Study 

In order to verify the validity of the proposed method of measuring emotions with an indirect question, we conducted a pilot study on a group of subjects from Poland: *N* = 152; 122 women, 29 men, and 1 person declared as non-binary; aged 18 to 54 years old (*M* = 24.13; *SD* = 7.54). The study was conducted in a between-subject model. One of the groups was asked to declare to which extent each of the 20 emotions causes a sense of threat among other people (indirect question), while the other group was asked to which extent each of the listed emotions causes a sense of threat in them (direct question). The measurements were done on a scale from 1 (“to a small extent”) to 100 (“to a large extent”), as in the first stage of the main study.

The between-subject *t*-test analysis revealed that the intensity of declared emotions was higher in the indirect condition (*M* = 61.73; *SD* = 12.65) than in the direct condition (*M* = 49.84; *SD* = 17.91): *t*(134.89) = 4.73, *p* < 0.001, *d* = 1.33. It is important to note that not only was the intensity of emotions higher in the indirect condition, in the direct condition the mean was very close to the mean value of the scale, and the standard deviation was much higher than in the indirect condition, which suggests that the variance of answers was greater in the direct than in the indirect condition. Results of the presented pilot study suggest that using the indirect question about emotions, we could expect more intense answers not suppressed by defence mechanisms. 

### 3.2. Negative Emotions: Main Study

The baseline value from stage 1 was compared in a series of *t*-tests with all three consecutive stages. The analyses revealed that the value from stage 1 was higher than all values from the subsequent stages. The level of negative emotions in stage 1 (*M* = 4.31, *SD* = 1.09) was higher than in stage 2 (*M* = 4.17, *SD* = 1.21): *t*(2197.78) = 3.02, *p* = 0.003; in stage 3 (*M* = 3.87, *SD* = 1.32): *t*(1759.47) = 8.62, *p* < 0.001; and in stage 4 (*M* = 3.75, *SD* = 1.35): *t*(1393.19) = 10.18, *p* < 0.001. Figure 2 presents mean values (with standard errors of measurement) acquired in the four consecutive research stages.

A repeated measures ANOVA was run for stages 2 to 4 (the same group of participants), which revealed that they differed in terms of negative emotion intensity: *F*(1.98, 1617.38) = 33.36, *p* < 0.001, *η_p_*^2^ = 0.04. Greenhouse–Geisser correction on degrees of freedom was used because according to the value of Mauchly’s test, the assumption of sphericity was not met: *W* = 0.99, *χ*^2^(2) = 19.04, *p* < 0.001. We ran simple contrast analyses for the differences between stages 2, 3, and 4 (included in the ANOVA analysis). According to the values of simple contrast, the mean value of negative emotions in stage 3 was significantly lower than the mean value for stage 2: *F*(1, 817) = 62.04, *p* < 0.001, *η_p_*^2^ = 0.07. In addition, the mean value of negative emotions in stage 4 was significantly lower than the mean value for stage 2: *F*(1, 817) = 5.30, *p* = 0.022, *η_p_*^2^ = 0.01. 

### 3.3. Sense of Threat at Different Distances

The effects regarding the sense of threat at different distances were analysed. We started with analysing the differences between the sense of threat at different distances using a repeated-measures ANOVA. We revealed the main effect of distance: *F*(1.43, 1781.12) = 428.14, *p* < 0.001, *η_p_*^2^ = 0.26 (degrees of freedom reported with Greenhouse–Geisser correction), *W* = 0.61, *χ*^2^(2) = 622.87, *p* < 0.001. Simple contrast analyses revealed that the value for Poland (*M* = 4.72, *SD* = 1.10) differed significantly from the values for Europe (*M* = 5.62, *SD* = 1.01)—*F*(1, 1242) = 590.38, *p* < 0.001, *η_p_*^2^ = 0.32—and the whole world (*M* = 5.37, *SD* = 1.02)—*F*(1, 1242) = 328.93, *p* < 0.001, *η_p_*^2^ = 0.21.

Subsequently, we analysed the differences between the sense of threat (in general and divided into different distances), particularly the differences between stage 1 and the three following stages. The general sense of threat was the mean from the assessments at all three distances. The analyses revealed that the general sense of threat was higher in the first stage than in all three following ones. The same pattern was observed for Poland, Europe, and the whole world. The means, SEMs, and exact results of comparisons between the measurements are presented in Table 2.

For stages 2 to 4, we conducted repeated measures ANOVA analyses. We observed the main effect of distance: *F*(1.42, 1160.08) = 99.21, *p* < 0.001, *η_p_*^2^ = 0.11. Sense of threat for Poland averaged between three stages (*M* = 4.08, *SD* = 1.34) differed significantly from the sense of threat in Europe (*M* = 4.42, *SEM* = 1.28)—*F*(1, 817) = 128.86, *p* < 0.001, *η_p_*^2^ = 0.14—and the whole world (*M* = 4.38, *SEM* = 1.29)—*F*(1, 817) = 95.89, *p* < 0.001, *η_p_*^2^ = 0.11. The sense of threat in Europe and worldwide did not differ significantly. We also observed the main effect of research stage: *F*(1.98, 1621.17) = 42.39, *p* < 0.001, *η_p_*^2^ = 0.05. The general sense of threat from stage 2 was significantly higher than in stage 3—*F*(1, 817) = 62.12, *p* < 0.001, *η_p_*^2^ = 0.07—and in stage 4—*F*(1, 817) = 59.88, *p* < 0.001, *η_p_*^2^ = 0.07. The mean sense of threat did not differ significantly between stages 3 and 4.

Finally, we observed the effect of interaction between distance and the stage of the study: *F*(3.10, 2531.66) = 15.91, *p* < 0.001, *η_p_*^2^ = 0.02. In general, the sense of threat for Poland was lower than for Europe and the whole world in all three stages. The sense of threat for Europe was lower than for the whole world only in stage 2. The sense of threat in stage 2 was higher than in stages 3 and 4 for all distances and did not differ between stages 3 and 4 at all three distances. Means, SDs, and the exact results of comparisons between the measurements are presented in Table 3. Figure 3 presents mean values (with standard errors of measurement) for all three distances in all four stages.

## 4. Discussion

The aim of this study was to identify the dynamics of people’s affective response to the threat situation generated by the first wave of the COVID-19 pandemic. Such knowledge gathered during the coronavirus pandemic, which has been going on for over two years in various parts of the world, may be of practical value. It provides the premises for a more effective response and development of a strategy of interactions, not only therapeutic but, above all, prophylactic. The psychological, social [45], and health [46,47] effects of the pandemic pose challenges on a scale comparable to those after war cataclysms.

We focused on recording two manifestations of affective reactions: the intensification of negative emotions and the subjective sense of danger among Poles, measured four times at intervals of an average month during the first wave of the pandemic. We considered the first period of confrontation with the real prospect of viral infection and the uncontrolled consequences of the disease course as diagnostic for the emergence of universal and automatic strategies for responding to a threatening situation. In turn, long-term exposure to threat made it reasonable to predict changes in the dynamics of the intensity of affective attitudes.

It was expected, according to the knowledge of emotion dynamics and data collected in Germany [30], Great Britain [32], and Austria [33], that the intensity of both negative emotions and the sense of threat would decline in subsequent months of the pandemic compared with the intensity recorded at the onset. We also expected that the distinguished status of one’s own place (close distance) would modify the assessments related to safety. It was expected that assessment of the pandemic situation as threatening should be lower with regard to the situation of the pandemic in Poland as compared with that in Europe and around the world.

An Internet-based study, with a scheme of repeated measures, quasi-experimental research, was conducted. We found that the intensity of negative emotions (Hypothesis 1) and subjective sense of threat (Hypothesis 2) decreased at each stage of our study. The highest intensity of both affective states and the sense of threat was observed at the very beginning of the pandemic. This could be understood as the affective response to a completely new and objectively threatening situation, which is also confirmed by research regarding self-regulation [48,49]. In each successive period, despite the pandemic measures objectively increasing (including in Poland), the intensity of negative emotions and sense of threat decreased. So, the initial affective responses to the pandemic appeared to indicate a realistic attitude among the Poles. The declared assessments of the intensity of negative emotions and the sense of threat reflected the automatic and adaptive role of affective states in the regulation of attitudes. In contrast, the decrease in the intensity of negative emotions and the assessment of the sense of threat in a pandemic situation in subsequent measurements can be interpreted by referring to the phenomenon of habituation [50,51], the essence of which is the reduction in the strength of response to stimuli due to their repeatability as a result of stopping the signal transmission process at the level of lower cerebral centres. Thus, repeated exposure to a pandemic situation is leading to a decline in the intensity of affective responses.

The revealed pattern of the dynamics of the intensity of affective states (their decrease compared to the initial level) allows us to look at this phenomenon from the perspective of adaptive cognitive distortions such as unrealistic optimism [52,53]. Prolonged, intense experience of negative emotions and a sense of threat is not only uncomfortable but, above all, devastating for the mental condition, increases the intensity of depression symptoms [16], burdens cognitive resources, and disorganises logical thinking. So, the sense of such biases is short-lived, transient, and of profoundly adaptive significance, because they increase well-being, contribute to mental and physical health, and support productivity and motivation [54]. They help individuals recover a sense of control, reduce anxiety, and minimise the strain on psychological resources.

Two factors (also present in the situation of the pandemic)—controllability of events [55,56,57,58] and personal involvement—predict the appearance of such a distortion. Controllability of events means that we know what we can do in order to have control over the occurrence of a given event [59,60]. Residents of Poland heard a lot about the coronavirus spreading in China, and then they gradually received information about its appearance in various European countries. Only after a significant period of time did they learn that it had also appeared in Poland. Additionally, people were informed that frequent handwashing and avoiding large groups of people could decrease the likelihood of infection. Moreover, politicians during the elections assured the public that the pandemic was on the retreat. The second factor is personal involvement: health is certainly of central importance to most people, and it favours biases, consisting of a conviction about little risk.

The greater the uncertainty, the more important it becomes to make predictions because there is no clear indication of how the situation will develop [61]. This also applies to a pandemic. The tendency to form optimistic beliefs is adaptive as long as it relates to beliefs. When optimism ceases to be a belief and begins to incline to an unrealistic perception of reality, the person may miss the objective indicators of threat [58,62]. Among the number of different phenomena normally grouped under the heading “optimism bias”, authors distinguished between unrealistic comparative optimism and unrealistic absolute optimism [63]. Unrealistic comparative optimism describes a situation in which people evaluate their own prospects as better than those of similar others. They expect that positive outcomes are more likely and negative outcomes are less likely to occur for oneself than for others. Unrealistic absolute optimism means that people’s risk assessment or negative states are unrealistically positive when compared to an objective criterion, such as an actuarial risk assessment or actual outcomes, such as danger of infection.

It seems that the latter manifestation of distortion constitutes an adequate framework for the interpretation of the decrease in the intensity of affective states. The decrease in the intensity of negative emotions and the sense of threat in the face of objectively increasing rates of disease and hospitalisation reflects the effect of unrealistic absolute optimism. This makes adaptive sense: a lower intensity of negative affect sanctions the perception of a pandemic situation as less threatening over time. The phenomenon is universal. The results of our research are consistent with others, indicating that this motivational and cognitive phenomenon not only exists but is also a specific social norm [64]. The non-accidental nature of the obtained data is also confirmed by their repeatability: the next three measurements systematically confirmed the effect. The regulatory role of habituation and distortions explaining the decrease in the intensity of negative emotions and the sense of threat is complementary. The former limits emotional arousal, while the latter modifies cognitive assessment. The role of both seems difficult to overstate, particularly during the first wave of the pandemic.

Another result confirmed that pandemic risk assessments were dependent on the proximity to the site. The smaller distance from the self was favourable to the assessment of the situation. In Poland, the situation was assessed as less threatening than in Europe and around the world. Such a pattern of results was maintained regardless of the passage of time and was present in the following months, despite the rapidly increasing number of cases. It was initially justified, as the situation in Poland during the first outbreak was less critical than in Western Europe; as time went on though, the gap was rapidly closing.

Certainly, the motivation to see one’s place of residence as safer than other locations is a manifestation of the desire to satisfy the need for security (which played its part in generating these subjective assessments). It seems that we may be dealing with a different form of positive illusion than in the case of emotions: unrealistic comparative optimism. In an emergency, people tend to ignore important data, which results in distortion. People evaluate their own prospects as better than those of similar others (or another specific reference group). In other words, they expect that positive outcomes are more likely and negative outcomes are less likely to occur for oneself than for others [63].

Regardless of the detailed form, positive illusions are taken to give rise to beliefs that are poorly supported by the evidence and that leave the person with a more positive outlook than is warranted. Both of these positive illusions can interact with one another. For example, if the subject feels that the place he is in is less threatening than others, it “justifies” the decrease in the intensity of negative emotions.

The obtained results are important for health experts and persons who are responsible for developing prevention strategies, especially in the pandemic period. Succumbing to unrealistic optimism influences behaviour during an epidemic threat [65]. People revealing strong unrealistic optimism believe that they are immune to infection and that they are not affected by the threat [66]. This makes them less likely to be vaccinated [67].

Unrealistic optimism made patients reluctant to undergo medical treatment [68] and encouraged passivity and neglect in the face of real threats [69], such as the risk of infection. From this perspective, inclinations to react with unrealistic optimism increase the probability of ignoring the restrictions recommended during the pandemic, increasing the tendency to engage in risky behaviour. The problem seems to be significant because the effects of unrealistic optimism observed in many social groups are associated with disregarding the threat, and this may lead to an increase in the number of COVID-19 cases around the world.

Despite the negative consequences of unrealistic optimism, in the context of a pandemic, it seems important for people to be “realistic” optimists and believe that they can help to avoid getting sick and expect a positive development of events. It is therefore worth taking actions that strengthen such optimism. It can increase safety motivation, promote mental well-being, and enhance life satisfaction [70,71].

It has to be mentioned as a limitation of this study that it was conducted with the use of online tools, which impact and modify the course of cognitive processes [72], leading to more automatic information processing. Reading becomes sketchier and more fragmentary, and the information acquired online tends to be more poorly remembered and less correctly understood [73]. Hence, irrespective of the objective possibilities of conducting the studies during the pandemic, it is worth paying attention to the effects that digital tools and digital context have on the quality of data collected and, consequently, on their interpretation and generalisation of results. The online procedure also limits the control and standardisation of research conditions. The applied online procedures (one-time access to the questionnaire), the indirect nature of variable measurements, and a pilot study, however, were used to increase the reliability of the responses. 

In conclusion, the results of the study show two main phenomena: the habituation of feeling threatened and negative emotions in time, as well as unrealistic optimism regarding the “self-group” in comparison to “others” or more distant groups. The optimism is advantageous as a defensive mechanism; however, it has to be confronted with the actual situation, as very high optimism may lead to unreasonable behaviour (e.g., rejecting sanitary rules or vaccination). The results of the study may be useful in planning media communication regarding crises, as well as in the clinical work treating the psychological consequences of the COVID-19 pandemic.

## Figures and Tables

**Figure 1 ijerph-19-13497-f001:**
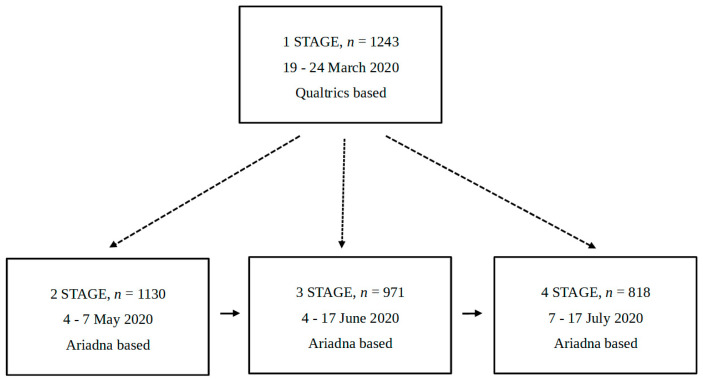
Research scheme used in the study.

**Figure 2 ijerph-19-13497-f002:**
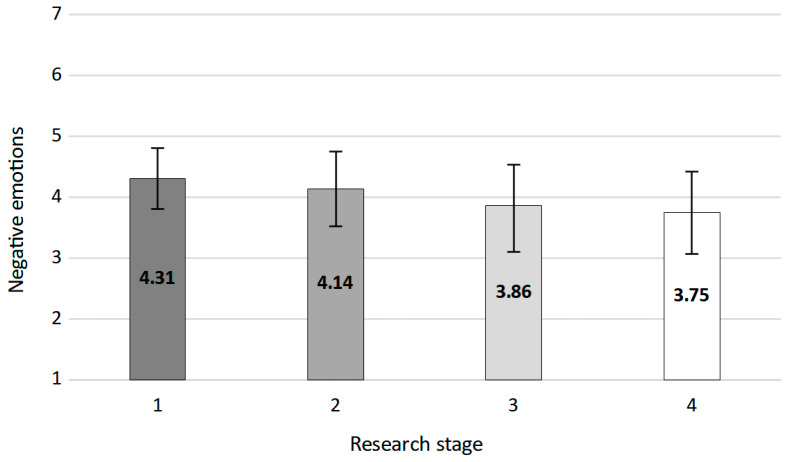
Intensity of negative emotions in four consecutive research stages. Error bars present standard deviations.

**Figure 3 ijerph-19-13497-f003:**
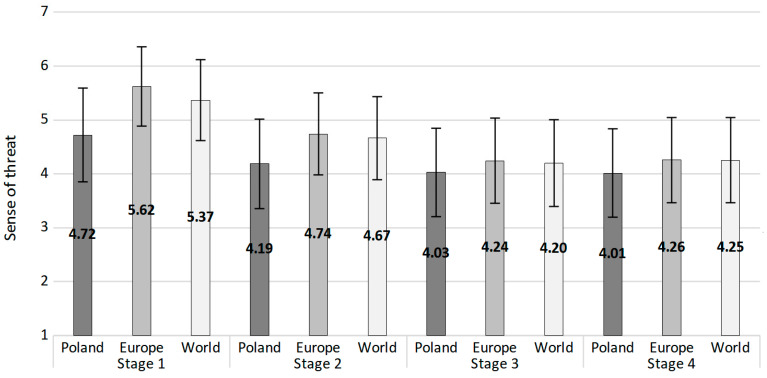
Mean values of sense of threat regarding Poland, Europe, and the whole world in four consecutive research stages. Error bars present standard deviations.

**Table 1 ijerph-19-13497-t001:** Number of participants in each stage divided by gender.

		First Stage	Second Stage	Third Stage	Fourth Stage
Participants	Women	1022 (82.2%)	569 (50.4%)	473 (48.7%)	387 (47.3%)
	Men	216 (17.4%)	561 (49.6%)	498 (51.3%)	431 (52.7%)
	Other	5 (0.4%)	-	-	-
	Total	1243	1130	971	818

**Table 2 ijerph-19-13497-t002:** *t*-test comparisons between stage 1 and the three following stages of the study. Means and SDs are presented next to the description of the measurement.

Research Stage		Stage 1			
	Measurement	Poland (*M* = 4.72, *SD* = 1.73)	Europe (*M* = 5.62, *SD* = 1.47)	World (*M* = 5.37, *SD* = 1.49)	General sense of threat (*M* = 5.24, *SD* = 1.43)
Stage 2	Poland (*M* = 4.12, *SD* = 1.66)	*t*(2371) = 8.61, *p* < 0.001			
	Europe (*M* = 4.70, *SD* = 1.52)		*t*(2331.93) = 14.96, *p* < 0.001		
	World (*M* = 4.62, *SD* = 1.53)			*t*(2371) = 12.10, *p* < 0.001	
	General sense of threat (*M* = 4.48, *SD* = 1.41)				*t*(2371) = 13.00, *p* < 0.001
Stage 3	Poland (*M* = 4.01, *SD* = 1.63)	*t*(2133.85) = 9.98, *p* < 0.001			
	Europe (*M* = 4.25, *SD* = 1.58)		*t*(2007.87) = 20.96, *p* < 0.001		
	World (*M* = 4.19, *SD* = 1.61)			*t*(2007.94) = 17.66, *p* < 0.001	
	General sense of threat (*M* = 4.15, *SD* = 1.48)				*t*(2212) = 17.56, *p* < 0.001
Stage 4	Poland (*M* = 4.01, *SD* = 1.63)	*t*(1814.62) = 9.40, *p* < 0.001			
	Europe (*M* = 4.26, *SD* = 1.57)		*t*(1664.93) = 19.77, *p* < 0.001		
	World (*M* = 4.25, *SD* = 1.57)			*t*(2059) = 16.24, *p* < 0.001	
	General sense of threat (*M* = 4.18, *SD* = 1.46)				*t*(2059) = 16.39, *p* < 0.001

**Table 3 ijerph-19-13497-t003:** *t*-test comparisons within the interaction of distance and study stage.

Research Stage		Stage 2			Stage 3			Stage 4	
	Measurement	Poland	Europe	Whole World	Poland	Europe	Whole World	Poland	Europe
Stage 2	Europe	*t*(1129) = -14.11, *p* < 0.001							
	Whole World	*t*(1129) = -11.72, *p* < 0.001	*t*(1129) = 4.11, *p* = 0.01						
Stage 3	Poland	*t*(970) = 3.14, *p* = 0.02							
	Europe		*t*(970) = 9.76, *p* < 0.001		*t*(970) = −6.16, *p* < 0.001				
	Whole World			*t*(970) = 9.22, *p* < 0.001	*t*(970) = −4.61, *p* < 0.001	*p* > 0.05			
Stage 4	Poland	*t*(817) = 2.98, *p* = 0.01			*p* > 0.05				
	Europe		*t*(817) = 9.12, *p* < 0.001			*p* > 0.05		*t*(817) = −5.63, *p* < 0.001	
	Whole World			*t*(817) = 7.66, *p* < 0.001			*p* > 0.05	*t*(817) = −5.47, *p* < 0.001	*p* > 0.05

## Data Availability

The data presented in this study are openly available in FigShare at https://doi.org/10.6084/m9.figshare.21354180.v1, accessed on 9 October 2022.

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
