# Peer review of "Affective Attitudes in the Face of the COVID-19 Pandemic: The Dynamics of Negative Emotions and a Sense of Threat in Poles in the First Wave of the Pandemic"

_ijerph, 2022, doi:10.3390/ijerph192013497_

Round 1

Reviewer 1 Report

As far as I am concerned the article is written keeping the academic standard, and it is well written; nonetheless, there are some points I think should be addressed: 

1.       I do not understand why there is strikethrough text on lines 166-167.

2.       Section 3.2: Although I agree with the statistical analysis the Authors carried out, I think that amount of numerical data could be in a table for better understanding, more didactical if you mean it.  

3.       References: Some of them seem to be written in different styles and fonts.

Author Response

Thank you for the review. We have revised the whole manuscript, changing a lot of details throughout the text. The changes are marked both with “tracked changes” function, as well as the yellow highlights. We hope that in general the changes improve the reception of the manuscript. The particular points brought in your review are addressed below:

  1. Thank you for pointing out this mistake, the text has undergone proofreading, which removed minor errors and typos, we also hope that it improved the general quality of English language used in the manuscript.
  2. Following your suggestion, we moved a large part of descriptive statistics as well as all the t-test comparisons to Tables 2. and 3. (section 3.3. in the current version of the manuscript). We also added a description of the results, presenting the general shape of the differences, without the exact statistics (section 3.3., paragraph 2 and 4). We hope that it improves the general reception of the Results section.
  3. The references section has also undergone proofreading, the font has been unified, we hope that we removed all the errors.

Reviewer 2 Report

Thank you for giving me the opportunity to review this manuscript. I have the following comments that I think the authors should consider addressing them before the manuscript is considered further for publication:

1. My major concern is that it is relatively difficult to gauge the objectives and methodology of the study. The objectives of the study are not clearly stated that could further direct on the methodology of the work. In fact, the paper is not reported or structured within the commonly acceptable guidelines for reporting research findings.

2. The study design is not clear. Authors mentioned four studies were conducted, but what are their designs, and objectives/hypothesis for those studies? It could be much appreciated if a flow diagram of the research methodology was conducted. Authors mentioned it was a web based study, further clarifications are required? 

3. Some of the results (for example the demographic characteristics of the participants) are placed in the methods part as well. These should be moved to the results section.

4. The authors need to describe how the questionnaire was developed. Was any content or face validity done? This is crucial as the paper deals with psychological attributes during pandemic times. Single-item questions without a validated scale score might not be appropriate.

5. The results were presented as mean in some figures with their standard errors. It would be more appropriate to present the measures of dispersion/95% confidence intervals.

6. Since it was an online survey, how was response veracity assured?

7. Overall, I feel that the authors need to restructure the paper, especially the methodology part in order for the readers to understand the significance and methodology of the study. Neither the abstract could enable readers to understand what and how was it done?

8. Please pay some attention to the language. I noticed multiple language errors and typos throughout the manuscript.

Reviewer 3 Report

This paper reports a very interesting reaction of the Polish people to the COVID-19 pandemic, which has become a global threat. This result is expected to have similar reactions not only in Poland but also in other countries. Please confirm only one point. The SD bars in the two figures (Figures 1 and 2) look the same. Is this SD bar notation and data correct? It seems that the SD bars will not be aligned so far, so please reconfirm.

Author Response

 Thank you for your review, we have revised the whole manuscript, changing a lot of details throughout the text. We hope that in general the changes improve the reception of the manuscript. The changes are marked both with “tracked changes'' function, as well as the yellow highlights. As for the error bars on the figures (Fig 2. and 3. in the current version of the manuscript), initially they were representing the standard errors of measurement (SEMs), that is why it was really difficult to distinguish the differences, as they were very short. It was also really difficult for our team to recognize whether the bars differ, so following your suggestion, as well as the suggestions of Reviewer 2, we changed the statistics representing errors to standard deviations (SDs). We hope that the current way of presenting statistics is more representative for the true dispersion of the data.

Round 2

Reviewer 2 Report

Thank you for the revisions and justifications. I am happy with the changes made. I look forward to see this paper soon in the scholarly literature.